# Chromosome-Wide Distribution and Characterization of H3K36me3 and H3K27Ac in the Marine Model Diatom *Phaeodactylum tricornutum*

**DOI:** 10.3390/plants12152852

**Published:** 2023-08-02

**Authors:** Yue Wu, Leila Tirichine

**Affiliations:** Nantes Université, CNRS, US2B, UMR 6286, F-44000 Nantes, France; yue.wu1@etu.univ-nantes.fr

**Keywords:** epigenetics, microalgae, evolution, post translational modifications of histones

## Abstract

Histone methylation and acetylation play a crucial role in response to developmental cues and environmental changes. Previously, we employed mass spectrometry to identify histone modifications such as H3K27ac and H3K36me3 in the model diatom *Phaeodactylum tricornutum*, which have been shown to be important for transcriptional activation in animal and plant species. To further investigate their evolutionary implications, we utilized chromatin immunoprecipitation followed by deep sequencing (ChIP-Seq) and explored their genome-wide distribution in *P. tricornutum*. Our study aimed to determine their role in transcriptional regulation of genes and transposable elements (TEs) and their co-occurrence with other histone marks. Our results revealed that H3K27ac and H3K36me3 were predominantly localized in promoters and genic regions indicating a high conservation pattern with studies of the same marks in plants and animals. Furthermore, we report the diversity of genes encoding H3 lysine 36 (H3K36) trimethylation–specific methyltransferase in microalgae leveraging diverse sequencing resources including the Marine Microbial Eukaryote Transcriptome Sequencing Project database (MMETSP). Our study expands the repertoire of epigenetic marks in a model microalga and provides valuable insights into the evolutionary context of epigenetic-mediated gene regulation. These findings shed light on the intricate interplay between histone modifications and gene expression in microalgae, contributing to our understanding of the broader epigenetic landscape in eukaryotic organisms.

## 1. Introduction

Diatoms, a group of unicellular photosynthetic organisms belonging to the stramenopile lineage, are characterized by their remarkable diversity, over 100,000 extant species known to date [1]. The evolutionary history of diatoms is closely linked to the biogeochemical cycles of the Earth’s oceans and atmosphere. As primary producers, diatoms play a pivotal role in the global carbon cycle, sequestering carbon dioxide and contributing significantly to marine primary productivity, thereby supporting complex aquatic food webs [2]. Their distinctive silica-frustule structure confers remarkable mechanical strength and resilience, enabling them to withstand predation and environmental stresses, and facilitating their adaptation to diverse aquatic environments [3]. With the advent of microscopy in the 19th century, diatoms were divided into mainly two groups based on the symmetry: centric diatoms, which are usually circular or triangular in shape and have radial symmetry, and pennate diatoms, which are elongated and have bilateral symmetry [4]. These two groups also differ in their ecological preferences and patterns of diversity. Within the two major groups, there are further sub-categories, such as polar centric and radial centric for centrics, and raphid, araphid for pennates. Phylogenetic studies using molecular markers, such as ribosomal RNA genes, have provided compelling evidence suggesting that pennate diatoms diverged from centric diatoms approximately 125 million years ago [5].

In the context of diatoms evolution, epigenetic mechanisms have likely played a role in shaping the diversity and adaptability of these organisms. Epigenetic studies in diatoms have only recently emerged within the past decade, following the release of two model diatom genomes, *Phaeodactylum tricornutum* and *Thalassiosira pseudonana* [6,7,8]. Current research suggests that the adaptive advantage of phytoplankton is potentially derived from epigenetic regulation mechanisms, rather than point mutation-based processes. This is because the latter may not occur at a sufficiently rapid rate to enable effective adaptation to the dynamic and rapidly changing conditions of the oceanic environment [6]. In eukaryotes, nuclear DNA is packaged in a highly organized chromatin structure that consists of repeating units called nucleosomes. Each nucleosome is composed of two H2A-H2B dimers and one (H3-H4)2 tetramer. The N-terminal tails of histone, project outward from the nucleosome core and undergo a range of various post-translational modifications, including methylation of Lysine (Lys) and Arginine, acetylation of Lys, phosphorylation of Serine and Threonine and ubiquitination of Lys [9,10].

Histone modifications play a critical role in regulating gene expression by modifying the structure of chromatin, thus affecting the accessibility of DNA to the transcriptional machinery. Additionally, these modifications can also recruit enzyme complexes to the chromatin, further influencing gene expression [11,12,13]. Histone acetylation typically leads to an open chromatin structure that is permissive to gene expression, while histone methylation can either activate or repress gene expression, depending on the site and extent of the modification [14]. For example, H3K4me3 is linked with transcription initiation, as it is typically found near the transcription start site (TSS) of actively transcribed genes and is associated with the recruitment of transcriptional machinery to the TSS. H3K9/14ac and H3K27ac are often present in promoter-associated nucleosomes at actively transcribed genes [15,16], and H3K36me3 plays a vital role in the regulation of transcriptional elongation by promoting efficient transcription following an elongating pol II [17]. Conversely, H3K27me3 has been linked to the inhibition of transcription elongation, suggesting its involvement in gene silencing [18]. 

The first genome-wide characterization of histone modifications in *P. tricornutum* was released in 2015, which identified 62 post-translational modifications (PTMs) on histones using mass spectrometry, including mono-, di-, and tri-methylation, acetylation, and mono-ubiquitination of lysines. Combining genome-wide mapping of six key histone marks (H3K4me2, H3K9me2, H3K9me3, H3K27me3 and H3K9/14ac, H3K4me3) with DNA methylation profiling revealed different chromatin states and gene expression patterns, extending the histone code to Stramenopiles. This epigenetic code displayed both unique and broadly conserved chromatin features, reflecting the chimeric nature of *P. tricornutum* genome [19,20,21]. 5mC (5-methylcytosine) is predominantly found in a CG context across repetitive sequences and transposable elements with low expression levels in diatoms, which is consistent with the repressive function of DNA methylation observed in other eukaryotes [22,23,24]. The conservation of the epigenetic machinery in diatoms offers a unique opportunity to gain insights into the processes underlying phenotypic plasticity and explore their role in evolutionary processes. A previous study revealed that PRC2 and its associated H3K27me3 mark play a critical role in the differentiation of cells in *P. tricornutum* [25]. The identification and characterization of these epigenetic marks offer a promising avenue for elucidating their biological roles and evolutionary significance, particularly in the context of *P. tricornutum*’s phylogenetic position. Such insights can provide valuable understanding of the underlying mechanisms that drive adaptation and survival in response to environmental changes. 

The dynamic regulation of histone modifications by a variety of histone modifying enzymes is essential for proper chromatin function and cellular homeostasis. The process of acetylation of ε-amino group in lysine on histone proteins is a crucial mechanism for the activation of transcription [26]. This modification is carried out by a group of proteins known as histone acetyltransferases (HATs), which can affect the structure and function of chromatin in a broad and non-specific manner. For example, p300/CBP-associated factor (PCAF) has been shown to catalyze preferentially acetylation of the ε-amino group of lysine 14, 18, and 27 in histone H3, which plays an important role in the remodeling of chromatin and the regulation of gene expression [27,28]. The regulation of H3K27 acetylation is complex and is often dynamically regulated by a balance between the activity of histone acetyltransferases (HATs) such as p300/CBP complex, PCAF and histone deacetylases (HDACs), which remove acetyl groups from histones [27,29,30]. 

In contrast, different histone methyltransferases (HMTs) are responsible for methylating different lysine or arginine residues on histones, and the effects of methylation can vary depending on the specific residue and degree of methylation. For example, the addition of the methyl group to the lysine residue at position 36 of histone H3 (H3K36) is catalyzed by histone methyltransferases, such as Set2 in yeast and “SET domain-containing protein 2” (SETD2) in humans [31]. H3K36me3 is an evolutionarily conserved histone modification that marks active transcription and is highly enriched at the transcribed regions of genes [32,33]. In yeast *Schizosaccharomyces pombe* and *Saccharomyces cerevisiae*, Set2 is the sole lysine methyltransferase that mediates all forms of H3K36 methylation, and couples H3K36me3 with transcriptional elongation [34]. SETD2, as the human orthologue of yeast Set2, is the primary methyltransferase catalyzing H3K36 trimethylation (H3K36me3) in vivo and interacts with hyperphosphorylated RNA polymerase II during transcription elongation [35]. The orthologous group of Set2/SETD2 and its lysine methyltransferase activity are also conserved across species [36]. In *Arabidopsis thaliana*, the homolog of SETD2, termed SDG8, has been identified as a critical enzyme involved in the di- and trimethylation of H3K36, while in *Drosophila melanogaster*, dSet2/HYPB is primarily responsible for H3K36me3 [37,38]. In *A. thaliana*, SDG8 was shown to exhibit specific regulatory functions in modulating histone methylation at H3K36 within gene bodies, particularly for genes associated with photosynthesis and metabolism that are responsive to light and/or carbon cues [39,40]. 

Previous genomic sequencing studies of *P. tricornutum* have revealed a comprehensive repertoire of histone-modifying enzymes, highlighting the remarkable conservation of these writers of histone modification marks in diatoms and their ancient evolutionary origins [21,41], including the identification of a protein encoding putative enzymes PtSETD2 (Phatr3_EG02211) for catalyzing the deposition of methylation on lysine 36 of histone H3 [41]. These findings greatly advanced our understanding of the epigenetic regulation in diatoms, and have provided important tools for further exploring the roles of epigenetics in diatom biology.

In this study, we used chromatin immunoprecipitation coupled with deep sequencing to investigate the distribution pattern and functional implications of H3K27ac and H3K36me3 in the model diatom *P. tricornutum*. Additionally, we sought to broaden our understanding of the H3K36me3 diversity in diatoms by analyzing the occurrence and evolution of H3K36me3-specific methyltransferase from other species using the Marine Microbial Eukaryote Transcriptome Sequencing Project database (MMETSP) [42]. Furthermore, we undertook a re-examination of the epigenetic code that was previously established in *P. tricornutum* [19,43,44] and analyzed the mapping profiles of H3K27ac and H3K36me3 in relation to other histone marks. Our aim was to determine their co-occurrence pattern and effect on transcriptional regulation of genes and transposable elements in this unicellular organism. 

## 2. Results & Discussion

### 2.1. Lineage Specific Features of SETD2 Homologs in Diatoms

To gain a better understanding of how Set2/SETD2 proteins have evolved to regulate gene expression and chromatin structure in different biological systems, we explored the orthologous group of Set2/SETD2 in unicellular organisms including different groups of microalgae. The orthologous groups of Set2/SETD2 protein show high sequence conservation across some main annotated functional domains (Figure 1A). They consist of triplicate AWS (associated with SET)-SET-PS (post-SET) motifs, responsible for the transfer of H3K36me [35], and a C-terminal Set2-Rpb1-interaction (SRI) domain that mediates the interaction with pol II [45]. Interestingly, the sequence analysis in InterPro suggests that MET-1, the homolog of Set2 in *Caenorhabditis elegans*, lacks an SRI domain, which may suggest that additional mechanisms are primarily responsible for recruiting Set2 to chromatin [46]. Profile hidden Markov models (HMMs) for SETD2 genes was built using multiple sequence alignment data obtained from other model species. The robustness of the HMMs was supported by high bootstrap values, indicating a significant degree of conservation among SETD2 homologs in these species (Appendix A). Subsequently, we scanned the profile against published diatom genomes including *P. tricornutum* and *T. pseudonana*, in addition to transcriptome libraries such as the MMETSP database [42] (refer to the “Materials and Methods” section) and seven other diatom transcriptomes, namely *Shionodiscus*, *Synedra*, *Pleurosigma*, *Navicula*, *Haslea*, *Guinardia*, and *Minidiscus,* that were sequenced by Genoscope. A maximum likelihood phylogenetic tree was then constructed to understand the evolutionary divergence of SETD2 among unicellular eukaryotes (Figure 1B). Interestingly, all sequences share three common conserved motifs (motif I, II, III) that match the AWS-SET-PS domain found in Set2/SETD2 through InterPro prediction, which provided insight into the functional and structural similarity of the SETD2 homologs across different diatom species. The presence of conserved domains motif IV, V, VI, VII in both polar centric diatoms and some raphid pennate diatoms, despite their inability to catalyze the H3K36me3 function, reveals a close relationship between these two groups and implies a shared evolutionary origin.

### 2.2. Chromosome-Wide Distribution of H3K27ac and H3K36me3 Histone Modifications in P. tricornutum

In addition to previous detection of H3K27Ac and H3K36me3 histone marks using mass spectrometry [19], we used western-blotting to further verify the existence of these marks in the genome of *P. tricornutum* and the specificity of the antibodies (Figure 1C). Then, to further characterize the genomic patterns associated with these two histone modifications on the new 25 to 25 telomere assembly of *P. tricornutum* [43,47], we performed Chromatin Immno-Precipitation (ChIP) on two independent biological replicates, using monoclonal antibodies specific to H3K27ac and H3K36me3, followed by DNA deep-sequencing (ChIP-Seq). Additionally, we used ChIP-qPCR to validate randomly selected loci, providing further evidence for the accuracy of H3K27ac and H3K36me3 ChIP-seq data (Figure 2A). Two biological replicates were sequenced for each mark and subsequent clustering analysis showed high correlation coefficients between replicates (Appendix A).

The H3K27ac modification covered nearly 13.24% of the *P. tricornutum* genome, with 4928 detected peaks with an average peak length of 747 bp (Figure 2C and Appendix A). Most H3K27ac-enriched nucleosomes were detected as sharp, narrow peaks (<1 kb) positioned near the transcription start sites (TSSs), which were mostly found at promoter regions (Figure 2B,D,E). We looked into the functional significance of H3K27Ac, and found that it is positively associated with active transcriptional activity. This is supported by our analysis of gene expression, which demonstrates that genes associated with H3K27Ac exhibit significantly higher expression levels compared to genes targeted by the Polycomb Repressive Complex 2 (PRC2) (wilcox.test, *p* < 0.01) (Figure 2F), which is known to repress gene expression [25,43]. Studies have shown that H3K27ac is enriched at active TSSs of genes, where it promotes the recruitment of transcriptional machinery to the promoter regions in other eukaryotes, such as mammals and rice [48,49,50]. These proteins, in turn, help initiate and maintain the transcription of the associated genes. On rare instances, there are few H3K27ac peaks (85 peaks) that appear to be localized in repeats (Appendix A), which mainly correspond to transposable elements (TE) relics or transposase genes. Among the identified TEs, 103 exhibited enrichments for H3K27ac. These particular TEs showed significantly higher expression levels compared to TEs marked with H3K27me3 which is a mark deposited by the PRC2 known to silence TEs in *P. tricornutum* (wilcox.test, *p* < 0.01) [19,43] (Figure 2G). This suggests the potential involvement of H3k27Ac in transcriptional activation of TEs. Furthermore, we categorized the TEs associated with H3K27ac and revealed a substantial enrichment of Class II TEs, particularly DNA transposons Mutator (DNA-DTM), which accounted for 34.95% (Figure 2H). Meanwhile, the TE category covered by PRC is mainly enriched with Class I TEs, which have long terminal repeats (LTRs), particularly in the copia (37.47%) and unknown (45.29%) superfamilies. Notably, the well characterized active DNA-DTM tends to land in regions of accessible chromatin with low methylation levels [51]. In contrast, many maize LTR elements are found inserted within other TEs, suggesting a preference for insertion into silenced chromatin [44]. Hollister et al. (2011) demonstrated in their study that small RNA-mediated silencing of LTR retrotransposons can induce changes in DNA methylation and histone modifications at nearby genes, resulting in altered gene expression [52]. This finding implies that certain TE categories have a tendency for acquiring H3K27ac marks and these TEs may be actively involved in regulatory processes, taking advantage of the accessible chromatin state. Overall, these results highlight the importance of TEs marked with H3K27ac in transcriptional regulation, and further research is needed to unravel the specific mechanisms by which these TEs affect the overall gene expression dynamics in *P. tricornutum*.

H3K36me3 was a relatively more abundant mark found in *P. tricornutum* genome (3400 detected peaks, 23.05% of the genome), forming approximately on average 2-kb-long blocks (Figure 2C). We found that trimethylated H3K36 was enriched throughout the promoter, coding sequences (CDS), peaking near the 3′ ends of transcription units (Figure 2B,D,E) and this pattern is highly conserved across most eukaryotes [53,54]. In accordance with its genomic distribution, H3K36me3 signals were sharply elevated after TSSs in active genes and showed a positive correlation with active transcription (Figure 2F). Notably, the levels of H3K36me3 were found to be significantly lower in genes marked by published repressive PRC2 (wilcox.test, *p* < 0.01). These results are consistent with the model that Set2 is recruited by the transcription elongation apparatus and that it methylates local nucleosomes during active transcription [55,56]. These findings will provide insight into the role of H3K36me3 in regulating gene expression and propose a potential mechanism for its specificity in transcriptional elongation in unicellular organisms. Furthermore, we identified 69 TEs showing enrichment for H3K36me3. Similar to H3K27ac, the expression levels of H3K36me3-marked TEs were found to be higher compared to the TEs covered by PRC (wilcox.test, 0.01 < *p* < 0.05) (Figure 2G). Notably, approximately 45% (31 out of 69) TEs were co-marked by both H3K36me3 and H3K27ac (Figure 2B, Appendix A), suggesting a potential functional interplay of TEs between these two histone modifications (Appendix A). Upon further investigation of TE categories associated with H3K36me3, we observed a significant enrichment of DNA transposons, particularly in the DNA/DTM (37.68%), DNA/DTC (27.54%), and DNA/Helitron (20.29%) categories (Figure 2H). It is known that Helitron transposons can capture gene fragments and insert themselves into gene-rich regions, potentially affecting gene expression [57,58].

### 2.3. Combinatorial Analysis of Histone Marks in P. tricornutum

To fully understand the function of a particular histone modification, it is essential to consider its interaction with other modifications, as per the ‘histone code hypothesis’. This hypothesis suggests that various histone modifications cooperate in a sequential, cooperative, antagonistic, or redundant way to establish and maintain a chromatin structure that is conducive to gene expression [59]. To investigate the relationship between H3K36me3, H3K27ac and previously characterized histone marks [19,43], we analyzed the co-marking patterns including six histone marks, namely H3K4me2, H3K4me3, H3K9/14Ac, H3K9me2, H3K9me3, H3K27me3, and DNA methylation (Appendix A). There are numerous possible combinations of histone marks and DNA methylation, mainly 40 most prevalent combination patterns are shown in Figure 3. To investigate the functional relationships between distinct histone marks and gene expression, we analyzed RNA-seq data and inspected the expression patterns of genes enriched with different histone marks. Our analysis revealed the presence of three primary chromatin states—active, repressive, and intermediate—based on the histone mark enrichments. The combination of H3K27ac, H3K36me3 with two other published active marks, H3K9_14ac and H3K4me3, co-localize on a greater number of genes (around 1800 genes) compared to other combinations (Figure 3 and Appendix A), leading to a permissive chromatin state, which is characterized by high levels of gene expression. The co-occurrence plot indicates that H3K27ac and/or H3K36me3 predominantly co-occurred with permissive histone marks, specifically H3K9_14ac, H3K4me3, and H3K4me2, are associated with active gene expression, and may work in a collaborative manner. This finding is consistent with previous studies of H3K36me3 in *S. cerevisiae*, which has been shown to play a role in regulating levels of H3K27ac and H3K56ac modifications, potentially preventing inappropriate transcriptional initiation by controlling nucleosome remodeling and histone exchange [60]. It is worth noting that this observed pattern is different from the antagonistic pattern of H3K36me-H3K27ac crosstalk that has been observed in fission yeast [61]. This highlights the importance of considering the specific system being studied and not generalizing findings across different organisms or experimental conditions, but further research is necessary to fully understand the underlying mechanisms in *P. tricornutum*. In addition, DNA methylation and three repressive histone marks, namely H3K27me3, H3K9me2, and H3K9me3, indicate a chromatin state that efficiently represses gene expression. Notably, this is a unique pattern that have not been observed in animals or plants. However, when these repressive histone marks are associated with H3K27ac and/or H3K36me3, they transition to an active chromatin state that can have a significant impact on the increase of global transcript levels, observed in few genomic locations (around 200 genes). By uncovering new patterns and relationships among various histone modifications, we can gain a more comprehensive understanding of how chromatin states contribute to gene expression and cellular differentiation.

### 2.4. Functional Analysis of H3K27ac and H3K36me3 in P. tricornutum

To gain insights into the functional categories enriched in H3K27ac marked genes, we further performed KEGG pathway enrichment analysis on H3K27ac/H3K36me3-associated genes (Figure 4). We observed H3K27ac mapping to genes involved in pathways related to cellular energy production, specifically oxidative phosphorylation and ATP synthesis. This indicates that these genes are actively transcribed and are likely playing a role in cellular energy metabolism, which is the process by which cells convert nutrients into energy in the form of ATP. Mitochondria are organelles that play a crucial role in this process, by integrating metabolic pathways to produce ATP through oxidative phosphorylation. Acetyl-CoA is an important molecule that fuels the mitochondrial TCA cycle and ETC for ATP production, while also serving as a substrate for certain enzymes called Histone Acetyl Transefrases (such as P300/CBP), which acetylate histones [62,63]. The statistical analysis revealed a strong association between the ribosome biogenesis pathway and H3K27ac annotated genes, with several ribosomal proteins being prominently featured in this pathway, suggesting that it plays a role in regulating their expression. Studies have found that H3K27ac is highly enriched at the promoter regions of ribosomal protein genes in mammalian cells, which is positively correlated with gene expression levels [64,65,66]. Further research may be needed to explore the mechanisms underlying this relationship and to determine its potential implications for cellular processes. 

Among the genes specifically targeted by H3K36me3, there are 144 loci involved in various metabolic pathways, including glucose metabolism, amino acid metabolism and TCA cycle. Recent studies have suggested that H3K36me3 acts as lipopolysaccharide-sensitive epigenetic mark that senses multiple upstream metabolic pathways to control the availability of S-adenosylmethionine (SAM), allowing direct communication between the metabolic state and the chromatin state [67]. This enables the cell to rapidly adapt to changes in the environment. In addition, H3K36me3-related genes were found to be involved in the regulation of nutrient and energy metabolism-associated genes in response to light, carbon, and nitrogen signals. Many of these genes are direct targets of the histone methyltransferase SDG8 [40,68], suggesting that epigenetic regulation of metabolic gene expression and metabolic enzyme activity play a critical role in stress responses and adaptation to changing environments.

## 3. Conclusions

Taken together, our findings suggest that the genes encoding H3 lysine 36 (H3K36) trimethylation-specific methyltransferase in diatoms exhibit both functional and structural similarities, providing a potential avenue to explore the evolution of Set2/SETD2 proteins in regulating gene expression and chromatin structure in microalgae. In *P. tricornutum*, H3K27ac and H3K36me3 are mainly located in genic regions and are associated with high-level of gene expression. Genes marked with these histone modifications have the highest expression levels and frequently co-occur with other permissive histone marks, such as H3K4me3 and H3K9/14ac. Additionally, KEGG pathway analysis revealed that the genes marked with H3K27ac and H3K36me3 are involved in various metabolic pathways that enable cells to adapt to environmental changes. Further investigations in *P. tricornutum* and other emerging unicellular models will certainly provide new insights and deepen our comprehension of the regulatory roles of these histone modifications in modulating gene expression and epigenetic regulation mechanisms in response to changing environments. Both marks are now available on the epigenome browser PhaeoEpiview [43], which represents a significant development in the study of epigenetic regulation in diatoms. This addition holds substantial promise for shedding new light on the ecological success of these fundamental organisms in modern oceans, while also enriching our understanding of the evolutionary history of epigenetic mechanisms in animals and plants. The unique chimeric origin of diatoms coupled to their position in the tree of life confer a specific significance to these insights.

## 4. Materials and Methods

### 4.1. Phylogenetic Analysis and Motif Analysis

A representative set of published reference sequences of H3K36me3-specific methyltransferase was selected, and a multiple sequence alignment (MSA) was generated using the MUSCLE alignment tool in the Geneious 11.0.5 software, then a maximum likelihood tree was constructed with default parameters. Next, a profile Hidden Markov Model (HMM) was built using HMMER 3.3.2 [69] to capture conserved sequence patterns and structural features of the protein family or domain. The HMM was then employed to search the MMETSP database for homologous sequences, and the results were filtered and curated to remove false positives and sequences with low similarity scores. To analyze motifs, the homologs search results were subjected to MEME-suite 5.5.3.org, an online software tool. The phylogenetic tree was constructed using MUSCLE 3.8.425 alignment tool in Geneious with default parameters and visualized via the iTOL server (http://itol.embl.de/ (accessed on 27 February 2023)).

### 4.2. Culture and Growth Conditions 

*Phaeodactylum tricornutum* Bohlin Clone Pt1 8.6 (CCMP2561) cells were obtained from the culture collection of the Provasoli-Guillard Marine Algae for Culture of Marine Phytoplankton (Bigelow Laboratory for Ocean Sciences, East Boothbay, ME 04544, USA.). Constantly shaken (100 rpm) cultures were cultivated at 19 °C, 70 µmol photons m^−2^ s^−1^ and with a 12 h light/12 h dark photoperiod in Enhanced Artificial Sea Water (EASW) [70] medium with a salinity of 21 g/L. For Chromatin immunoprecipitation-sequencing, cultures were seeded in duplicate at 50.000 cells/mL and grown side by side in 1000 mL Erlenmeyer flasks until early-exponential at 1,000,000 cells/mL. Culture growth was measured using a hematocytometer (Fisher Scientific, Pittsburgh, PA, USA).

### 4.3. Chromatin Extraction and Immunoprecipitation

Chromatin isolation was performed as described previously [71] with a few modifications. Chromatin extractions were made from 1 L cultures until cell density reaches around 1 million cells/mL. Culture was centrifuged and quickly washed with phosphate-buffered saline (PBS), then cross-linked with 1% formaldehyde for 15 min. Fixation was stopped by addition of 0.125M Glycine. Cross-linked culture was centrifuged and washed with PBS buffer then centrifuged with pellets and stored in −80 °C. Aliquots of 50 mL culture pellets were resuspended in 5 mL of Extraction buffer I (0.4 M sucrose, 1 mini tablet Roche per 50 mL, 10 mM MgCl_2_, 5 mM 2- mercaptothanol, 10 mM Tris-HCl pH8) and kept on ice for 20 min. After centrifugation (4000 rpm, 20 min, 4 °C), pellets were resuspended in 1 mL of Extraction Buffer II (0.25 M sucrose, 10 mM Tris-HCl pH8, 10 mM MgCl_2_, 1% triton, 1 mini tablet Roche diluted in 1 mL (for 50 mL), 5 mM 2-MERCAPTOETHANOL) and centrifuged (4000 rpm, 20 min, 4 °C). This step was repeated until the pellet became almost white. Pellet was resuspended in 300 μL of Extraction Buffer III (1.7 M sucrose, 1 minitablet diluted in 1 mL (for 50 mL), 0.15% Triton X-100, 2 mM MgCl_2_, 5 mM 2-MERCAPTOETHANOL 10 mM Tris HCl pH 8). The 300 μL resuspended pellet were added to a clean Eppendorf tube with 300 μL buffer III and centrifuged (13,000 rpm, 60 min, 4 °C). After removal of the supernatant, the chromatin pellet was resuspended in 300 μL of Nuclei Lysis Buffer (50 mM Tris HCl pH 8, 10 mM EDTA, 1 mini tablet of protease inhibitors diluted in 1 mL (for 50 mL), 1% SDS). Each ChIP-seq experiment was conducted in two independent biological replicates. Then chromatin immunoprecipitation was performed as described previously [71].

### 4.4. ChIP-qPCR

Chromatin Immunoprecipitation (ChIP) was conducted as described previously [71]. DNA concentrations were quantified with the Qubit dsDNA BR Assay Kit (Thermo Fisher Scientific, Waltham, MA, USA). Quantitative PCR experiments were performed using LightCycler DNA Master SYBR Green Mix on the Biorad LightCycler (CFX96 Touch Real-Time PCR System, Instruments 1855196, Bio-Rad Laboratories, Inc., Hercules, CA, USA). IP enrichments were assayed with the following protocol: 1 μL of immunoprecipitated DNA samples (IP), input and mock DNA were mixed with 5 μL LightCyclerW DNA Master SYBR Green I 2X, 3 μL forward/reverse primers (1 μM), and 1 μL H2O. For each IP sample, the enrichment of certain histone modification on specific loci was calculated with the following equation:%Enrichment level = 100/2 ^(Cq[ChIP]-(Cq[Input]-Log2(Input Dilution Factor))^(1)

We performed ChIP-qPCR validation using monoclonal antibodies against H3K37ac and H3K36me3. Primers designed on randomly selected genes and used for ChIP-qPCR validation were listed in Appendix A.

### 4.5. Western Blot Analysis

Western blots were performed on nuclear extracts prepared from 50 mL cultures containing approximately 1 million cells/mL. Extracted nuclei were lysed in 20 μL Nuclei Lysis Buffer (50 mM Tris HCl pH 8, 10 mM EDTA, 1 mini tablet of protease inhibitors diluted in 1 mL (for 50 mL), 1% SDS), as described previously [71]. The protein concentration in the nuclear extracts was determined using the BCA Protein Assay kit. Subsequently, 50 μg and 100 μg of nuclear extracts were loaded on a 15% SDS polyacrylamide gel. Membranes were probed using the following high affinity CHIP-grade monoclonal antibodies: H3K27ac (Cell Signaling Technology D5E4) and H3K36me3 (Cell Signaling Technology D5A7).

### 4.6. Co-Occurrence and Correlation Analysis

ChIP-seq peaks were annotated with gene annotation from lifted genome annotation [43]. The intersection of different sets of genes overlapping with individual ChIP-seq peaks was performed using UpSetR 1.4.0 [72]. All combinations of overlap are derived and ordered from most to least overlapping categories. Non-overlapping peak sets are removed from the plot. Replicate correlation was performed using multiBigwigSummary tools from deepTools 3.5.1.

### 4.7. ChIP-Seq Analysis and Expression Analysis

Pair-end sequencing of H3K27ac and H3K36me3 ChIP and input samples was performed on Illumina NovaSeq platform with read length of 2 × 150 bp. Previously published ChIP sequencing for H3K9me2, H3K9me3, H3K4me2, H3K27me3, H3K9/K14Ac and H3K4me3 were retrieved from NCBI’s Gene Expression Omnibus accessions GSE68513 and GSE139676 [19,25]. Raw reads were filtered and low-quality read pairs were discarded using Trim Galore 0.6.7 (https://doi.org/10.5281/zenodo.5127899 (accessed on 23 July 2021)) with a read quality (Phred score) cutoff of 20 and a stringency value of 3bp. Using the 25 to 25 telomere assembly published in 2021 as a reference genome, the filtered reads were mapped using Bowtie2 2.4.5 [73]. We then performed the processing and filtering of the alignments using Samtools 1.15 “fixmate -m” and “markdup -r” modules [74]. Two biological replicates for each ChIP were performed and read counts showed a good Pearson correlation by Deeptools multiBamSummary v3.5.1 with a bin size of 1000 bp [75].

To identify regions that were significantly enriched, we used MACS2 v2.2.7.1[76] on the combination of the two replicates with “callpeak --qvalue 0.05 --nomodel --SPMR --bdg” options. In addition, extension size was set to the arithmetic mean of the two IP replicates fragment size for each mark, as determined by MACS2 predicted module with “-m 2 70” MFOLD value. Furthermore, we used MACS2 with calling mode “--broad” to call peaks for H3K37ac and H3K36me3. Output normalized Fold Enrichment signal files were generated with MACS2 v2.2.7.1 “bdgcmp” module and transformed to BigWig using Deeptools bedGraphToBigWig. Then, Pearson correlation between two biological replicates for H3K27ac and H3K36me3 was performed using Deeptools 3.5.1 plotCorrelation. Further peak annotations and comparisons were done using ChIPseeker v1.32.1 [77]. For genome-wide visualization of peaks, we used PhaeoEpiView (https://PhaeoEpiView.univ-nantes.fr (accessed on 23 May 2023)). All graphs were created using ggplot2 v3.4.0 [78].

### 4.8. KEGG Analysis

We performed KEGG analysis on a set of genes that were enriched in epigenetic regulatory marks, using the DAVID 2021 software (Database for Annotation, Visualization, and Integrated Discovery) [79]. Our target functional categories were directed pathways derived from “Pathways Chart” in DAVID. All the directed terms were extracted with the corresponding enrichment score values. A smaller EASE score indicates a higher degree of significance, meaning that the gene set is more enriched in the functional category. In our study, the cutoff for statistical significance is an EASE score of 0.05, which corresponds to a false discovery rate (FDR) of 5%.

## Figures and Tables

**Figure 1 plants-12-02852-f001:**
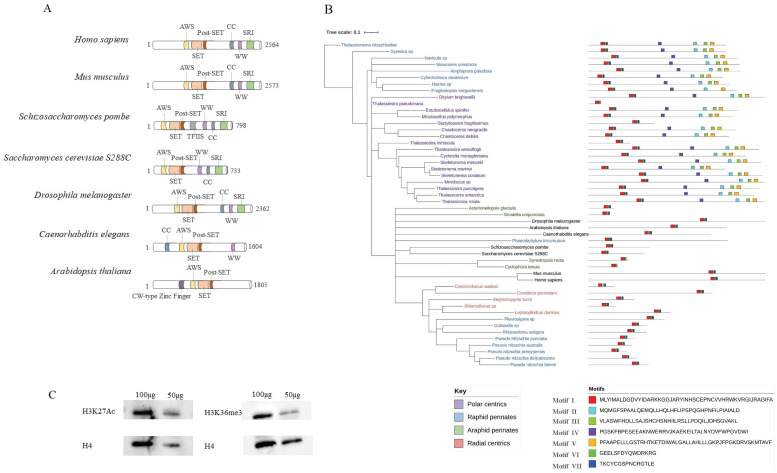
**Homolog of the histone methyltransferases Setd2.** (**A**) Domain structure of histone methyltransferases Setd2 in model species; (**B**) Phylogenetic analysis of Setd2 histone methyltransferase homologs in diatoms from MMETSP database; (**C**) Western blot analysis using monoclonal antibodies against H3K27ac and H3K36me3 in *P. tricornutum*, with histone H4 serving as a loading control.

**Figure 2 plants-12-02852-f002:**
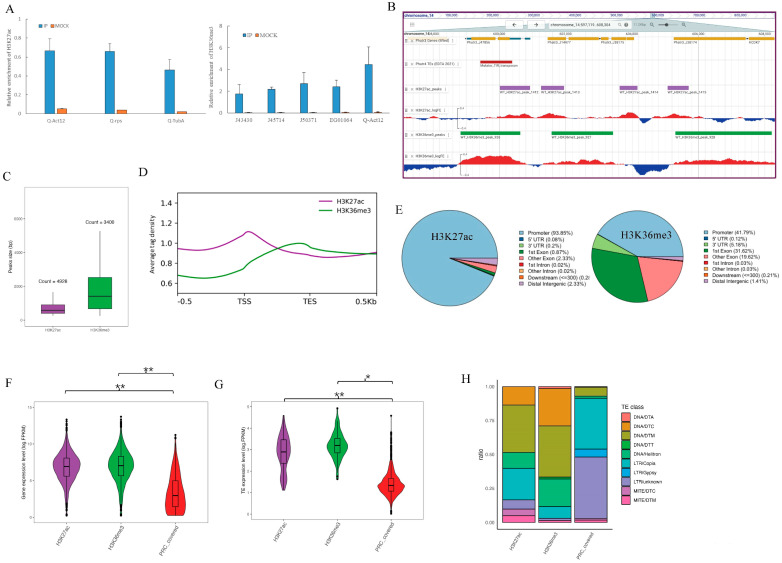
**Genome-wide distribution of H3K27ac and H3K36me3 in *P. tricornutum.*** (**A**) ChIP−qPCR validation of randomly chosen loci for H3K27ac and H3K36me3. (**B**) Snapshot of the PhaeoEpiview browser illustrating the tracks of genes (yellow), transposable elements (red). The ChIP-seq patterns display the distribution of H3K27ac (purple) and H3K36me3 (green) on chromosomes 14. The MACS2 detected peaks and log2 fold enrichment between the IP and input are displayed. (**C**) Peak (detected by MACS2) size distribution of H3K27ac (purple) and H3K36me3 (green). The box is delimited, bottom to top, by the first quartile, the median, and the third quartile, and the number above each box represents the number of peaks detected. Outliers are not shown. (**D**) Enrichment profile of H3K27ac and H3K36me3 along genes (upstream TSS, coding region, downstream TES). The average tag density is the number of sequence reads per gene. (**E**) Repartition of H3K27ac and H3K36me3 on annotated genome features. The pie charts represent the proportion of peaks that overlap with the genomic features indicated in the legend panel. (**F**) Expression levels of genes marked by H3K27ac and H3K36me3, respectively. Gene expression was inferred from the calculated fragments per kilobase of transcript per million mapped reads (FPKM) values. If a *p*-value is less than 0.05, it is flagged with one star (*). If a *p*-value is less than 0.01, it is flagged with 2 stars (**). (**G**) Expression levels of TEs marked by H3K27ac, H3K36me3, and PRC (polycomb repressive complex), respectively. TE expression was inferred from the calculated fragments per kilobase of transcript per million mapped reads (FPKM) values. (**H**) Proportion of TE families covered by H3K27ac, H3K36me3, and PRC.

**Figure 3 plants-12-02852-f003:**
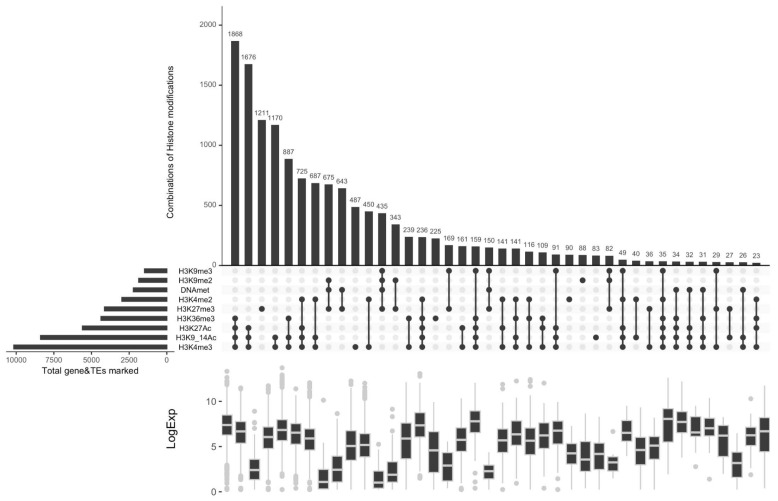
**Co-occurrence analysis of epigenetic marks in *P. tricornutum.*** For each set of intersections, a black-filled circle is drawn. The size of the intersections is depicted in the vertical bar. The vertical black line connects the different datasets that intersect. The number of genes and TEs for each chromatin state are shown on the top of the bars. Gene expression levels corresponding to each chromatin state are displayed at the bottom.

**Figure 4 plants-12-02852-f004:**
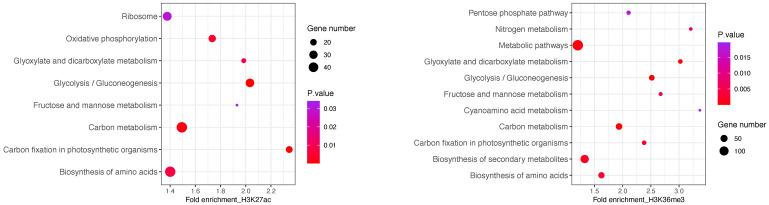
**KEGG Analysis of H3K27ac and H3K36me3 in *P. tricornutum.*** Scatter plot of KEGG pathways based on H3K27ac (**left**) and H3K36me3 (**right**) fold enrichment. Each pathway is represented by a circle, with the size of the circle indicating the number of genes associated with that pathway. The color of each circle represents the *p*-value of the pathway. The legend for the *p*-value color scale is shown on the right side of the figure.

## Data Availability

The datasets generated and analyzed in the present study can be accessed via BioProject PRJNA971863.

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
