# Peer review of "Chromosome-Wide Distribution and Characterization of H3K36me3 and H3K27Ac in the Marine Model Diatom Phaeodactylum tricornutum"

_plants, 2023, doi:10.3390/plants12152852_

Round 1

Reviewer 1 Report

The authors report the presence of H3K27ac and H3K36me3 marks in Phaeodactylum tricornutum. They analyzed the methyltransferase specific to H3K36me3 and provided KEGG enrichment analysis of genes that are potentially regulated by this mark.

Although the authors extensively explain what diatoms are and discuss epigenetic regulation in general, they fail to provide an introduction specific to the topic at hand: methyltransferases and these particular marks in Phaeodactylum and diatoms as well as Stramenopiles. Instead, most of the attention is given to what is known in yeast, plants, and animals. It is worth noting that the authors themselves are experts in the field and have contributed significantly to the literature on the epigenetic regulation of Phaeodactylum.

This omission makes it difficult to assess the study's contribution to the field and determine what is novel. Consequently, it is challenging to answer most of the questions regarding the significance of the findings. I suggest the authors amend this issue and provide a discussion on the relevance of their findings in relation to the existing literature.

For instance, the study by Hoquin et al. (2023), titled "The model diatom Phaeodactylum tricornutum provides insights into the diversity and function of microeukaryotic DNA methyltransferases" in Communications Biology, is not cited and should be considered. Additionally, the authors should include references such as Zhao X, Deton Cabanillas AF, Veluchamy A, Bowler C, Vieira FRJ, and Tirichine L (2020) in "Probing the Diversity of Polycomb and Trithorax Proteins in Cultured and Environmentally Sampled Microalgae" published in Frontiers in Marine Science, and Zhao X, Rastogi A, Deton Cabanillas AF, Ait Mohamed O, Cantrel C, Lombard B, Murik O, Genovesio A, Bowler C, Bouyer D, et al.'s study "Genome-wide natural variation of H3K27me3 selectively marks genes predicted to be important for cell differentiation in Phaeodactylum tricornutum" published in New Phytologist (2021, 229:3208-3220). These are only some references I could quickly find. 

By incorporating additional references such as these and discussing their findings in light of the existing literature, the authors can better establish the relevance of their research. At present, I cannot make any claims on for example the novelty of the data or interest to readers. In any case, the literature study should not be left to the reader. 

The english is very good. Only some small typos were found. 

Author Response

Reiewer 1 :

Comments and Suggestions for Authors

The authors report the presence of H3K27ac and H3K36me3 marks in Phaeodactylum tricornutum. They analyzed the methyltransferase specific to H3K36me3 and provided KEGG enrichment analysis of genes that are potentially regulated by this mark.

Although the authors extensively explain what diatoms are and discuss epigenetic regulation in general, they fail to provide an introduction specific to the topic at hand: methyltransferases and these particular marks in Phaeodactylum and diatoms as well as Stramenopiles. Instead, most of the attention is given to what is known in yeast, plants, and animals. It is worth noting that the authors themselves are experts in the field and have contributed significantly to the literature on the epigenetic regulation of Phaeodactylum.

This omission makes it difficult to assess the study's contribution to the field and determine what is novel. Consequently, it is challenging to answer most of the questions regarding the significance of the findings. I suggest the authors amend this issue and provide a discussion on the relevance of their findings in relation to the existing literature.

For instance, the study by Hoquin et al. (2023), titled "The model diatom Phaeodactylum tricornutum provides insights into the diversity and function of microeukaryotic DNA methyltransferases" in Communications Biology, is not cited and should be considered. Additionally, the authors should include references such as Zhao X, Deton Cabanillas AF, Veluchamy A, Bowler C, Vieira FRJ, and Tirichine L (2020) in "Probing the Diversity of Polycomb and Trithorax Proteins in Cultured and Environmentally Sampled Microalgae" published in Frontiers in Marine Science, and Zhao X, Rastogi A, Deton Cabanillas AF, Ait Mohamed O, Cantrel C, Lombard B, Murik O, Genovesio A, Bowler C, Bouyer D, et al.'s study "Genome-wide natural variation of H3K27me3 selectively marks genes predicted to be important for cell differentiation in Phaeodactylum tricornutum" published in New Phytologist (2021, 229:3208-3220). These are only some references I could quickly find. 

By incorporating additional references such as these and discussing their findings in light of the existing literature, the authors can better establish the relevance of their research. At present, I cannot make any claims on for example the novelty of the data or interest to readers. In any case, the literature study should not be left to the reader. 

Response: We thank the reviewer for the remark. We have now included a concise overview of the epigenetic history about Phaeodactylum, diatoms, and Stramenopiles in the latest version of the manuscript and have cited key publications.

Comments on the Quality of English Language

The english is very good. Only some small typos were found. 

Reviewer 2 Report

In this manuscript, Wu and Tirichine present a detailed characterisation to markers of histone modification in the model diatom species of Phaedoctylum tricornutum, based on their distribution in the chromosomes, impact on transcription, combinatorial impact with other markers on gene expression, and their potential targets of gene functions. They also present a phylogenetic analysis of a key histone methyltransferase to assess conservation of distinct sequence features among diatoms, relative those from model eukaryotic organisms.

The manuscript is generally well-written. The Introduction is comprehensive, and the methodology appears scientifically sound. The results are interesting, and I believe that this work is a useful contribution to the research community. It is also refreshing to see how far the diatom research has progressed, compared to the other eukaryotic microbes!

However, some key information appears missing from this submission, thus affecting my capability to adequately review the manuscript. Importantly, captions for all figures and tables are not provided (or at least I cannot find them), which affects my ability to really understand these items. The three cited supplementary figures are also missing in the submission system, so I could not review these items. I attempted this review based on the information I have and provide more-detailed comments below. Line numbers follow the generated PDF.

General comment about the presentation of results:

The introduction is well written, and it sets the expectation about the structure of the results (in L112-122): (A) distribution pattern and (B) functional implications of the two markers, (C) evolution of the methyltransferase, then (D) co-occurrence analysis of the two markers with other histone markers. In the R&D section, results of (C) are presented first, followed by (A), then (D), ending with (B). This presentation makes the flow of the text a little disjointed, and the expectations set up in the Introduction were not met. The first paragraph of the R&D section (L125-140) is essentially background information, which may be better presented as discussion of (not background for) the results after results are presented or incorporated in the Introduction. I recommend the authors to consider revising the text so the readers would get to the results earlier on in the R&D section, and revise the last paragraph of Introduction (L112-122) to better reflect the outline of the results and improve the flow.  

Other comments:

1. L117-122: a wordy sentence - consider simplifying or breaking it down.

2. L127: "cerevisiaes" -> "cerevisiae"

3. L133-134: This sentence seems redundant, because the next sentence (L134-135) repeats the same information. 

4. Figure 1A: the labels of AWS and CC for the Drosophila sequence appear misplaced and not aligned with the lines pointing to the features

5. Figure 1B: can you provide a scale bar in amino acid residues for the diagram of protein sequences shown next to the tree? Given the variable sequence lengths, are these sequences shown in full-length, or they could be fragmented due the nature of the generated data? For clarity, the motif labels of I through VII used in the text should be defined also in the motifs legend.

6. L174: by "25 to 25 telomere assembly", did you mean telomere-to-telomere genome assembly that consists of 25 chromosomes? This instance is spotted again in L409. Feel free to ignore this comment if this is indeed a common term in the field. I can't seem to find such information during my brief online research.

7. L182 and L185: Figure 2B is cited in both instances, but it is not clear how it shows the relevant information about the 4928 peaks or the narrow peaks near TSSs. 2B shows a zoomed in region of Chromosome 14 - I assume this is an example, but it is not clear from the current text without further elaboration or a figure caption. 

8. 185-187: It is not clear how Figure 2F shows a positive association of H3K27Ac with active transcriptional activity. 

9. L191-192: Table S1 is expected to show information about repeat sequences at which the "85" peaks are localized. It is not clear which information in Table S1 relates to. The 103 TEs that exhibited enrichments for H3K27ac (L193) are indicated in this table instead. The lack of table legends/description is affecting clarity of the information presented.

10. L196: from what test was this p value derived? This should be noted clearly, e.g. "xxxx test, p = xxxx".

11. L218: same issue with Figure 2B as above, and the directionality of the CDS is a little obscured, so the peaks near 3'-end of transcription units are not clear. There seem to be a few highlights here, which may be clarified by citing the specific figure panel for each piece of information, rather than citing all three panels together - they show rather distinct information.

12. L220-221: same comment as above for Figure 2F. How can we discern a positive correlation from the figure?

13. L225-226: Table S1 should be cited here for the 69 TEs.

14. L227: from what test was this p value derived? 

15. L229: which info in Figure 2B relate to the 45% TEs? I suspect this refers to Figure S2B instead, but I cannot verify this.

16. L252: rather than stating "around 1800 genes", noting the exact number (1868) would be clearer so the readers can find the info quickly from Figure 3. Also, without a caption, it is not clear what the LogExp graph below the UpSet plot is showing.

17. L331-332: which substitution model was used? This info is important for maximum likelihood inference.

18. L335-336: "and the results were filtered and curated to remove false positives and sequences with low similarity scores" - how exactly were they filtered, and what score threshold was used?

19. L342-343: This the outdated name for the collection centre that was renamed to NCMA in 2011.

20. L358, 361, 365 (and maybe elsewhere): mercaptothanol should not be in all caps. Also, I note inconsistencies in the capitalisation of L for litre, e.g. mL (L357) versus ml (L360)

21. The capitalisation style of the subheadings (e.g. 4.5, 4.6 and 4.7) is inconsistent.

22. In my quick glimpse of the reference list - reference 1 is incomplete, reference 42 has HTML tags in the title.  

Author Response

Reviewer 2 :

Comments and Suggestions for Authors

In this manuscript, Wu and Tirichine present a detailed characterisation to markers of histone modification in the model diatom species of Phaedoctylum tricornutum, based on their distribution in the chromosomes, impact on transcription, combinatorial impact with other markers on gene expression, and their potential targets of gene functions. They also present a phylogenetic analysis of a key histone methyltransferase to assess conservation of distinct sequence features among diatoms, relative those from model eukaryotic organisms.

The manuscript is generally well-written. The Introduction is comprehensive, and the methodology appears scientifically sound. The results are interesting, and I believe that this work is a useful contribution to the research community. It is also refreshing to see how far the diatom research has progressed, compared to the other eukaryotic microbes!

However, some key information appears missing from this submission, thus affecting my capability to adequately review the manuscript. Importantly, captions for all figures and tables are not provided (or at least I cannot find them), which affects my ability to really understand these items. The three cited supplementary figures are also missing in the submission system, so I could not review these items. I attempted this review based on the information I have and provide more-detailed comments below. Line numbers follow the generated PDF.

In response to the reviewer comment, we have now provided detailed captions for all figures and tables in the revised manuscript. We hope these additions will improve the clarity and comprehension of the manuscript. We have thoroughly reviewed the submission system, and it appears that there was a technical issue during the submission process that resulted in the absence of the cited supplementary figures. To rectify this, we have re-uploaded the supplementary figures, and they are now accessible for your review. We are grateful for the reviewer efforts in reviewing the manuscript based on the available information

General comment about the presentation of results:

The introduction is well written, and it sets the expectation about the structure of the results (in L112-122): (A) distribution pattern and (B) functional implications of the two markers, (C) evolution of the methyltransferase, then (D) co-occurrence analysis of the two markers with other histone markers. In the R&D section, results of (C) are presented first, followed by (A), then (D), ending with (B). This presentation makes the flow of the text a little disjointed, and the expectations set up in the Introduction were not met. The first paragraph of the R&D section (L125-140) is essentially background information, which may be better presented as discussion of (not background for) the results after results are presented or incorporated in the Introduction. I recommend the authors to consider revising the text so the readers would get to the results earlier on in the R&D section, and revise the last paragraph of Introduction (L112-122) to better reflect the outline of the results and improve the flow.  

Response: We thank the reviewer for bringing up this point. We have made the necessary adjustments by relocating the paragraph from the R&D section to the introduction. Specifically, we placed it immediately after introducing H3K36me. This change has improved the manuscript's flow and overall coherence.

Other comments:

  1. L117-122: a wordy sentence - consider simplifying or breaking it down.

Response: We thank the reviewer for this remark. , We have split the statement into two sentences, hoping that it is easier to read.

  1. L127: "cerevisiaes" -> "cerevisiae"

Response: That was corrected.

  1. L133-134: This sentence seems redundant, because the next sentence (L134-135) repeats the same information. 

Response: We have made the necessary adjustments.

  1. Figure 1A: the labels of AWS and CC for the Drosophila sequence appear misplaced and not aligned with the lines pointing to the features

Response: We have made the necessary corrections.

  1. Figure 1B: can you provide a scale bar in amino acid residues for the diagram of protein sequences shown next to the tree? Given the variable sequence lengths, are these sequences shown in full-length, or they could be fragmented due the nature of the generated data? For clarity, the motif labels of I through VII used in the text should be defined also in the motifs legend.

Response:  The sequences have been included in their full length, and this has been specified in the legend. Moreover, we have provided explanations in the legend regarding the meaning of each of the domains (I to VII) referred to in the sequences.

  1. L174: by "25 to 25 telomere assembly", did you mean telomere-to-telomere genome assembly that consists of 25 chromosomes? This instance is spotted again in L409. Feel free to ignore this comment if this is indeed a common term in the field. I can't seem to find such information during my brief online research.

Response: . Yes, "25 to 25 telomere assembly" refers to a telomere-to-telomere genome assembly consisting of 25 chromosomes. In 2008, the genome assembly of P. tricornutum consisted of 33 scaffolds, but it was updated to 25 chromosomes in 2022. To provide a clear illustration, we mentioned this in our manuscript.

  1. L182 and L185: Figure 2B is cited in both instances, but it is not clear how it shows the relevant information about the 4928 peaks or the narrow peaks near TSSs. 2B shows a zoomed in region of Chromosome 14 - I assume this is an example, but it is not clear from the current text without further elaboration or a figure caption. 

Response: Thanks. We corrected the corresponding manuscript and please review the revised version. We have removed the citation of Figure 2B in the section discussing the information about the 4928 peaks. Figure 2B illustrated that most H3K27ac peaks are annotated by gene regions, which  correspond to these peaks that have a sharp increase near TSS regions where they promotes the recruitment of transcriptional machinery.

Chromosome 14 as an example. We have revised the relevant section accordingly. Additionally, we would like to clarify that the top of Figure 2B shows the region size, which can be used to estimate the size of some peaks <1kb, although it may not be entirely precise. However, for a comprehensive statistical analysis of all detected peaks, please refer to Figure 2C.

  1. 185-187: It is not clear how Figure 2F shows a positive association of H3K27Ac with active transcriptional activity. 

Reponses: We thank the reviewer for the feedback. In order to more vividly exhibit the positive association of H3K27Ac with active transcriptional activity, we have updated new Figure 2F and included the expression of genes covered by the Polycom complex for comparison. Additionally, Figure 3 has been added to provide a more comprehensive exhibition of the relationship between H3K27Ac and other marks. This figure will help to illustrate the findings and enhance the understanding of the positive association between H3K27Ac and active transcriptional activity.

  1. L191-192: Table S1 is expected to show information about repeat sequences at which the "85" peaks are localized. It is not clear which information in Table S1 relates to. The 103 TEs that exhibited enrichments for H3K27ac (L193) are indicated in this table instead. The lack of table legends/description is affecting clarity of the information presented.

Response: We apologize for the confusion. To clarify, based on our analysis, we identified 4928 peaks through H3K27ac, out of which 85 peaks were found to be annotated by 105 TEs. In P. tricornutum, certain gene regions overlap with regions occupied by TEs. Therefore, it is possible that the annotation of these 85 peaks includes a small proportion that is annotated by both the gene regions and the corresponding TEs. To provide a comprehensive overview, we have included a Table S1 that displays the total number of peaks marked by H3K27ac, as well as the number of peaks annotated by genes and TEs, respectively.

  1. L196: from what test was this p value derived? This should be noted clearly, e.g. "xxxx test, p = xxxx".

Response: We thank the reviewer for pointing this out. It is wilcox.test. We have made the necessary corrections.

  1. L218: same issue with Figure 2B as above, and the directionality of the CDS is a little obscured, so the peaks near 3'-end of transcription units are not clear. There seem to be a few highlights here, which may be clarified by citing the specific figure panel for each piece of information, rather than citing all three panels together - they show rather distinct information.

Response: We thank the reviewer for the feedback. We have made the necessary revisions to provide more details in the figure legend. Additionally, we have included arrows to indicate the directionality of the CDS for each gene displayed in Figure 2B. This will help in visualizing the orientation of the CDS regions within the genes.

  1. L220-221: same comment as above for Figure 2F. How can we discern a positive correlation from the figure?

Response:  We have addressed this issue and provided additional details in the response above

  1. L225-226: Table S1 should be cited here for the 69 TEs.

Response: Table S1 is now cited.

  1. L227: from what test was this p value derived? 

Response:  We have made the necessary correction and included the wilcox.test as part of the analysis.

  1. L229: which info in Figure 2B relate to the 45% TEs? I suspect this refers to Figure S2B instead, but I cannot verify this.

Reponse: Figure 2B was included to illustrate a case where one TE is co-marked by both H3K36me3 and H3K27ac. This is visible from the presence of the red track corresponding to the Mutator_TIR_transposon on Figure 2B.

  1. L252: rather than stating "around 1800 genes", noting the exact number (1868) would be clearer so the readers can find the info quickly from Figure 3. Also, without a caption, it is not clear what the LogExp graph below the UpSet plot is showing.

Response: We thank the reviewer for bringing that to our attention. We have given more explanation in the legend, like “The gene expression levels corresponding to each chromatin state are displayed at the bottom.”

  1. L331-332: which substitution model was used? This info is important for maximum likelihood inference.

Response: We explained in the L152-155. Profile hidden Markov models (HMMs) for SETD2 genes was built using multiple sequence alignment data obtained from other model species.

  1. L335-336: "and the results were filtered and curated to remove false positives and sequences with low similarity scores" - how exactly were they filtered, and what score threshold was used?

Response: The hmmsearch algorithm was used to search sequence databases MMETSP for sequence homologs: The e-value threshold was set to 1e-10, indicating a stringent significance cutoff; The "best_hit_score_edge" and "best_hit_overhang" were set to 0.05 and 0.25, respectively, which determine the scoring edges and overhangs for identifying the best hit; The percent identity cutoff was set to 50%, indicating a minimum sequence similarity threshold.

  1. L342-343: This the outdated name for the collection centre that was renamed to NCMA in 2011.

Response: We thank the reviewer for the updated information and we have made the necessary corrections accordingly.

  1. L358, 361, 365 (and maybe elsewhere): mercaptothanol should not be in all caps. Also, I note inconsistencies in the capitalisation of L for litre, e.g. mL (L357) versus ml (L360)

Response:  We thank the reviewer for the remark. We have made the necessary corrections.

  1. The capitalisation style of the subheadings (e.g. 4.5, 4.6 and 4.7) is inconsistent.

Response:  We have made the necessary corrections

  1. In my quick glimpse of the reference list - reference 1 is incomplete, reference 42 has HTML tags in the title.  

Response:  We thank the reviewer for bringing that to our attention. We have made the necessary corrections

Reviewer 3 Report

The acetylation- and methylation-mediated post-modification of histone in the model diatom has been examined. Based on the data collected from the website and the western blot analysis of the acetylation of the chromosome of P. tricornutum, the role of acetylation and methylation of histones for the regulation of gene expression is identified. Several genes upon such regulatory machinery are proposed according to the previous published papers.

The data are solid.

It can be accepted for publication in the present form.

Author Response

Reviewer 3:

Comments and Suggestions for Authors

The acetylation- and methylation-mediated post-modification of histone in the model diatom has been examined. Based on the data collected from the website and the western blot analysis of the acetylation of the chromosome of P. tricornutum, the role of acetylation and methylation of histones for the regulation of gene expression is identified. Several genes upon such regulatory machinery are proposed according to the previous published papers.

The data are solid.

It can be accepted for publication in the present form.

We thank the reviewer for his/her comment and the appreciation of our work